Earth System
Open Access Science Discussions
Data

# A High-Resolution Calving Front Data Product for Marine-Terminating Glaciers in Svalbard

Tian Li[1,2], Konrad Heidler[2], Lichao Mou[2], Ádám Ignéczi[1], Xiao Xiang Zhu[2,3], Jonathan L. Bamber[1,2]

[1] Bristol Glaciology Centre, School of Geographical Sciences, University of Bristol, Bristol, BS8 1SS, UK
[2] Chair of Data Science in Earth Observation, Department of Aerospace and Geodesy, Technical University of Munich, Munich 80333, Germany
[3] Munich Center for Machine Learning, Technical University of Munich, Munich 80333, Germany

*Correspondence to*: Tian Li (tian.li@bristol.ac.uk)

**Abstract.** The mass loss of glaciers outside the polar ice sheets has been accelerating during the past several decades and has
been contributing to global sea-level rise. However, many of the mechanisms of this mass loss process are not well understood, especially the calving dynamics of marine-terminating glaciers, in part due to a lack of high-resolution calving front observations. Svalbard is an ideal site to study the climate sensitivity of glaciers as it is a region that has been undergoing amplified climate variability in both space and time compared to the global mean. Here we present a new high-resolution calving front dataset of 149 marine-terminating glaciers in Svalbard, comprising 124919 glacier calving front positions during
the period of 1985-2023 (https://doi.org/10.5281/zenodo.8399899) (Li et al., 2023). This dataset was generated using a novel automated deep learning framework and multiple optical and SAR satellite images from Landsat, Terra-ASTER, Sentinel-2, and Sentinel-1 satellite missions. The overall calving front mapping uncertainty across Svalbard is 46 ± 21 m. The newly derived calving front dataset agrees well with recent decadal calving front observations between 2000 and 2020 (Kochtitzky and Copland, 2022) and an annual calving front dataset between 2008 and 2022 (Moholdt et al., 2022). The $R^2$ of the glacier
calving front change rates between our product and the latter is 0.98, indicating an excellent match. Using this new calving front dataset, we identified widespread calving front retreats during the past three decades, across most regions in Svalbard except for a handful of glaciers draining the ice caps Vestfonna and Austfonna on Nordaustlandet. In addition, we identified complex patterns of glacier surging events overlaid with seasonal calving cycles. These data and findings provide insights into understanding glacier calving mechanisms and drivers. This new dataset can help improve estimates of glacier frontal ablation
as a component of the integrated mass balance of marine-terminating glaciers.





## 1. Introduction

Glaciers and ice caps (GIC) distinct from the Greenland and Antarctic ice sheets are a significant contributor to global sea-
level rise in addition to thermal expansion (Intergovernmental Panel on Climate Change, 2023; Meredith et al., 2019). Their
mass loss has been accelerating during the early twenty-first century and their thinning rates have doubled (Hugonnet et al.,
2021). Specifically, the mass loss from Arctic glaciers during 2006-2015 contributed to sea-level rise at a similar rate ($0.6 \pm 0.1\ mm/yr$) as the Greenland Ice Sheet in response to the accelerated warming trend in the Arctic (Intergovernmental Panel
on Climate Change, 2023). Recent observations show that the maximum warming rate on Earth ($> 1.25°C/10yr$) during
1979-2021 lies in the Russian Arctic close to Svalbard (Rantanen et al., 2022), which is one of the most climatically sensitive
regions in the world (van Pelt et al., 2018; Serreze and Barry, 2011).

Svalbard is an Arctic Archipelago located near the northeast coast of Greenland and lies close to the northern limit of warm
North Atlantic water (Nuth et al., 2010). Its climate displays extreme variability in both space and time. The southwest region
has milder and more humid conditions while the northeast is colder and drier (Schuler et al., 2020), making it an ideal region
to study the response of glaciers to climatic forcing. In Svalbard, the warming rate is $1.7°C/10yr$ since 1991, about seven
times the global average (Nordli et al., 2020). Glaciers on Svalbard have been losing mass since the 1960s with a trend towards
more negative mass balance since 2000 (Schuler et al., 2020; Nuth et al., 2010). High-resolution regional climate models reveal
that modest atmospheric warming in the mid-1980s forced the limit of the firn zone (the boundary between ice and compacted
snow) to the hypsometric peak, leading to firn cover reduction, albedo reduction, and increased surface runoff, amplifying the
mass loss from all elevations (Noël et al., 2020). By linking historical and modern glacier observations, it was predicted that
the twenty-first century glacier thinning rates in Svalbard would more than double the rates from 1936-2010, with a strong
dependence on air temperature (Geyman et al, 2022).

Despite recent progress in estimating the mass balance of glaciers in Svalbard, uncertainties remain, especially the
quantification of frontal ablation – a combination of calving and basal melting. Frontal ablation is a key component of the total
mass balance of marine-terminating glaciers, with the other being the climatic mass balance (Schuler et al., 2020). Despite its
importance, most global glacier models do not include the frontal ablation component at all (Rounce et al., 2023). In Svalbard,
15% of the glaciers are marine-terminating and in other Arctic sectors it is significantly higher (Oppenheimer et al., 2019).
They account for about 60% of the total glacierized area (Błaszczyk et al., 2009) and experienced one of the highest frontal
ablation rates in the northern hemisphere. However, there have only been two systematic studies estimating the frontal ablation
of glaciers in Svalbard (Błaszczyk et al., 2009; Kochtitzky et al., 2022). Błaszczyk et al. (2009) estimated the frontal ablation
rates of 163 Svalbard tidewater glaciers during a short period from 2000-2006. Kochtitzky et al. (2022) updated this record by
estimating the frontal ablation with a decadal time resolution for 2000-2010 and 2010-2020.

One major limitation of frontal ablation estimates is the scarcity in calving front observations of marine-terminating glaciers
(Kochtitzky et al., 2023), which is essential for determining the relative contributions of calving and submarine melting
(Schuler et al., 2020) and their governing processes. A detailed understanding of the calving mechanism and its drivers is
crucial for the accurate prediction of glacier response to future climate forcing and consequent sea level change (Benn et al.,
2007; Kochtitzky et al., 2023). The currently available calving front datasets for marine-terminating glaciers in Svalbard are
either limited to a small sample of glaciers (Murray et al., 2015; Strozzi et al., 2017; Holmes et al., 2019; Nuth et al., 2019) or
low temporal resolutions in calving front observations (Błaszczyk et al., 2009; Carr et al., 2017; Kochtitzky and Copland, 2022;
Nuth et al., 2013; Moholdt et al., 2022).

Calving front mapping of glaciers beyond the Greenland Ice Sheet has primarily relied on manual delineation from optical
satellite imagery such as Landsat and ASTER (McNabb and Hock, 2014; Kochtitzky and Copland, 2022; Cook et al., 2019).



This often results in low spatial coverage and temporal resolution, as optical images are often influenced by the presence of clouds and the polar night. With the availability of new optical satellite missions such as Sentinel-2 and Landsat-9, as well as the SAR satellite Sentinel-1, it is possible to achieve a short image acquisition interval of 1 to 3 days. In the meantime, the growing availability of extensive satellite catalogue imposes a challenge for manual delineation. There is, therefore, a need for efficient automated methods. In recent years, deep learning has demonstrated promising capabilities in accurately mapping glacier calving fronts (Mohajerani et al., 2019a; Cheng et al., 2021; Heidler et al., 2022; Loebel et al., 2023; Gourmelon et al., 2022; Zhang et al., 2019; Baumhoer et al., 2019, 2023). Mohajerani et al. (2019) pioneered the application of deep learning in glacier calving front mapping by developing a U-Net architecture to isolate the calving front from satellite images, the method was tested on Helheim Glacier in Greenland with a mean deviation of 96.3 m from ground truth which is manually mapped calving front from Landsat images. Building on this, Heidler et al. (2022) proposed a novel deep learning framework HED-UNet by combing semantic segmentation and edge detection, which outperforms the traditional U-Net framework. So far, these deep learning frameworks have only been applied to a small sample of glaciers mainly located on the Greenland and Antarctic Ice Sheets. Nonetheless, these case studies serve as a foundation for automated, high-temporal-resolution mapping of glacier terminus locations on a large spatial scale for glaciers outside the ice sheets.

Here, we introduce a novel automated processing pipeline designed to map glacier calving fronts using a new deep learning framework Charting Outlines by Recurrent Adaptation (COBRA), which outperforms image segmentation models by combining convolutional neural networks and active contour models for calving front mapping (Heidler et al., 2023). This study yields a new high-resolution glacier calving front data product containing 124919 calving front traces for 149 marine-terminating glaciers in Svalbard during the period of 1985-2023 (Li et al., 2023), utilizing data from multiple optical and SAR satellite sensors, including Landsat, ASTER, Sentinel-2 and Sentinel-1. This newly compiled dataset offers unprecedented temporal density, that is valuable for analysing both the seasonal and interannual variations in glacier calving fronts, as well as capturing surge events.

## 2. Data and Methodology

### 2.1 Automated satellite image downloading from Google Earth Engine

To generate the calving front data product, optical images from three different satellite platforms Landsat, Terra-ASTER and Sentinel-2, along with SAR images in the Extra Wide (EW) swath mode from Sentinel-1, spaning the time period from 1972 to January 2023, were used. The satellite images were acquired from the Google Earth Engine (GEE) platform with a diverse range of image resolutions, repeat cycles and operation durations shown in Table 1. The detailed workflow for downloading satellite images automatically for marine-terminating glaciers from different GEE satellite image collections (Table A1) is shown in Figure 1.

Our selection of glaciers in Svalbard is based on the tidewater glacier terminus data product generated by Kochtitzky and Copland (2022) which includes areal change polygons for all marine-terminating glaciers across the Arctic in two different time periods: 2000-2010 and 2010-2020. To begin, the Kochtitzky and Copland (2022) frontal areal change polygons of each glacier were used to produce the glacier domain shapefiles (Box 1 in Figure 1). For each glacier, all the available different areal change polygons generated in two different time periods were first merged into one single polygon. Then the Minimum Bounding Rectangle (MBR) of this merged polygon was generated. The final glacier domain polygon (black boxes in Figure 2a) was produced by adding a 1500-m buffer length to the MBR. If the final glacier domain polygon contains multiple polygons likely to be associated with tributary glaciers, these polygons were then divided into separate individual glacier area change polygons and assigned unique identifiers by adding sequential letters to its original Randolph Glacier Inventory (RGI) version 6 glacier id (RGI Consortium, 2017) as new a glacier id, this updated glacier id was used throughout the whole study.



In total, we generated 220 glacier domain shapefiles (hereinafter referred to as 220 marine-terminating glaciers - we took tributary glacier as an independent glacier) (black boxes in Figure 2). The domain shapefile was used in defining the glacier

spatial extent to be used in querying satellite images from the GEE API.

For each glacier domain, satellite images were retrieved from four distinct satellite platforms, namely Landsat 1-9, Terra-ASTER, Sentinel-2A/B and Sentinel-1A/B (Table 1). The images were downloaded throughout the entire time span of each satellite mission and were used in mapping the glacier calving front locations. For optical satellite images downloaded from Landsat, Terra-ASTER and Sentinel-2, we set a cloud filter threshold of 40%. Furthermore, a universal threshold for a non-

data pixel ratio per image is set as 50% for both optical and SAR images. If the proportion of non-data pixels in a given satellite image exceeds 50%, it is presumed that this image may lack a sufficient number of pixels for accurate predictions. For the 220 marine-terminating glacier domains in Svalbard, 1135074 satellite images were downloaded for the glacier calving front prediction over the period of 1972-2023 in our study.

**Table 1. The image resolutions of different satellite sensors used in the calving front mapping.**

| Satellite Platform | Resolution | Availability | Repeat Cycle | Band |
|---|---|---|---|---|
| Landsat | 30 m | 1972 | 16 days | Near-infrared band |
| ASTER | 30 m | 2000 | 16 days | Near-infrared band |
| Sentinel-2 | 10 m | 2015 | 10 days | Near-infrared band |
| Sentinel-1 | 40 m | 2014 | 12 days | HH band (EW mode) |


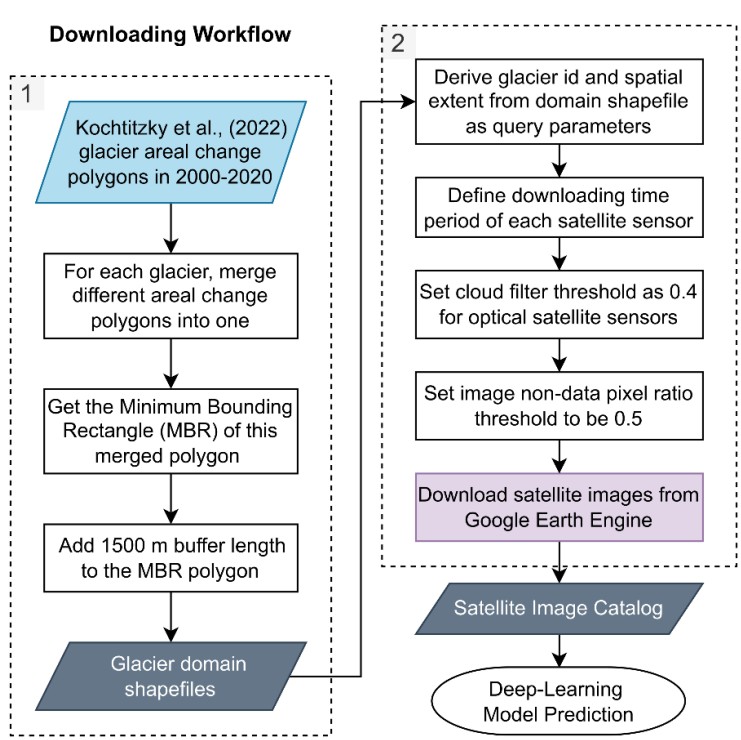

**Figure 1. The workflow of generating glacier domain shapefiles (box 1) and automated downloading satellite images from Google Earth Engine (GEE) (box 2) for Svalbard marine-terminating glaciers.**

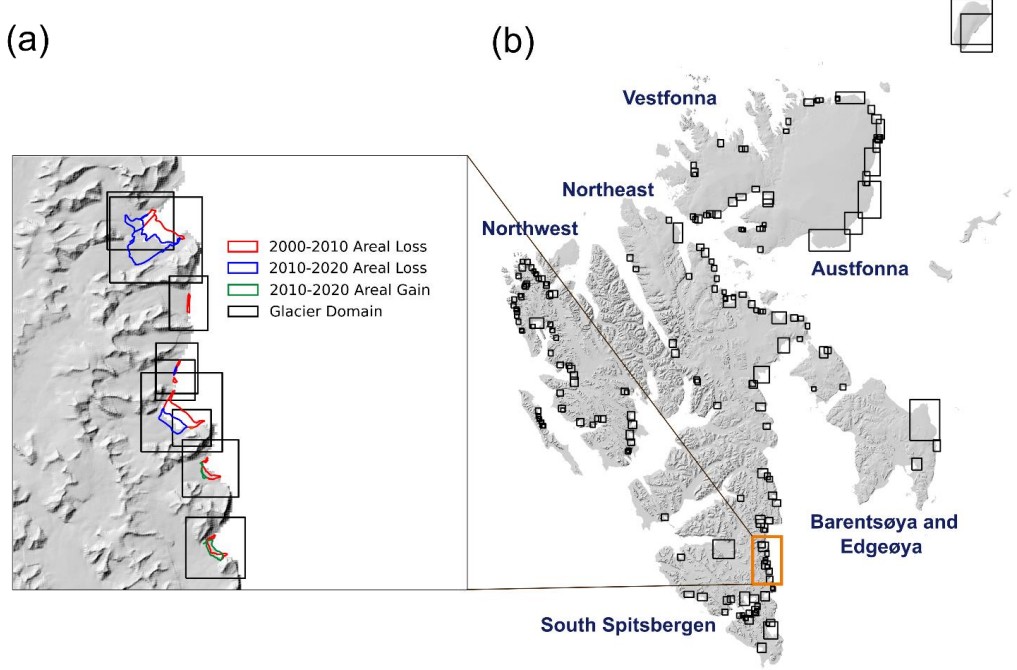

**Figure 2. a) Examples of glacier areal change polygons (colored outlines) generated in Kochtitzky and Copland. (2022) and the glacier domain polygons derived in this study (black boxes). The glacier areal loss in 2000-2010 was denoted as red polygon, the glacier areal loss in 2010-2020 was denoted as blue polygon, and the glacier areal gain in 2010-2020 was denoted as green polygon. b) The spatial distributions of 220 glacier domains generated in this study (black boxes), orange box denotes the zoomed-in region shown in subplot (a).**

**2.2 Deep learning model and preprocessing**

We used a deep learning model Charting Outlines by Recurrent Adaptation (COBRA) to predict the glacier calving front locations. The COBRA model combines a convolutional neural network (CNN) for feature extraction and an active contour model for the delineation (Heidler et al., 2023). Unlike the traditional image segmentation model such as CALFIN (Cheng et al., 2021) and HED-UNet (Heidler et al., 2022) which separate an image into land-ice and ocean classes, the COBRA model

can directly output the calving front line segment as a shapefile instead of recovering the vectorized contour from intermediate predictions in a semantic segmentation approach. Figure 3a shows the model architecture, and it comprises two different components: a backbone and a prediction head. The backbone of the COBRA architecture utilizes a versatile two-dimensional CNN to extract meaningful semantic features from the input imagery, here the Xception backbone was employed (Chollet, 2016; Cheng et al., 2021). The second component consists of a prediction head known as the "snake head" which leverages

the feature map of the backbone to generate the ultimate network predictions. The snake head starts with an initial calving front contour with vertices generated in the center of the image, then progressively refines the contour by incorporating sampled values from the feature map extracted from the backbone network and iterating this process for four times (Heidler et al., 2023).



The loss function of the COBRA model is based on the Dynamic Time Warping (DTW) loss which measures the similarity between the predicted contour and the true contour (Heidler et al., 2023), the loss function is shown as Equation (1)

$$\mathcal{L}_{DTW}(p,t) = \min_{(i_k,j_k)_{k\in[K]}\in\kappa} \sum_k \|p_{ik} - t_{ik}\|_2^2 \tag{1}$$


where the predicted contour p is represented by vertices $p_i$ with $1 \leq i \leq V$, and the true contour t is represented by vertices $t_j$ with $1 \leq j \leq V$. $\kappa$ denotes the set of all possible realignments $(i_k, j_k)_{k\in[K]}$ that satisfy the following three conditions: 1) For any $i \in \{1, ..., V\}$ there is a k with $i_k = i$; 2) For any $j \in \{1, ..., V\}$ there is a k with $j_k = j$; 3) The sequences $i_k$ and $j_k$ are non-decreasing in k.

The model was trained for 500 epochs on the CALFIN training dataset (Cheng et al., 2021) which includes 1541 Landsat optical images and 232 Sentinel-1 SAR images for 66 Greenlandic glaciers in 1972-2019. In addition, it was tested on three different test sets including the CALFIN test set, the TU Dresden (TUD) (Loebel et al., 2022) which includes 1127 Landsat optical images in 2013-2021 for 23 glaciers, as well as the Baumhoer dataset (Baumhoer et al., 2019) which includes 62 Sentinel-1 SAR images for glaciers located in Antarctica. The Adam optimizer (Kingma and Ba, 2014) with an initial learning 155 rate of $10^{-3}$ was used in the training process. The model was implemented in JAX using the Haiku framework (Heidler et al., 2023).

By the time of COBRA model development, CALFIN is the most complete glacier calving front mapping training dataset available for the Northern Hemisphere (Cheng et al., 2021). Although the sizes of tidewater glaciers in Greenland are typically much larger than those in Svalbard, their geomorphological characteristics are similar (Benn et al., 2007). Therefore, we used 160 this pretrained COBRA model to map glacier calving fronts in Svalbard. In order to maintain the consistency with the training dataset (Cheng et al., 2021), the near-infrared band of the optical images and the HH band of the SAR images were used (Table 1). Each satellite image was initially cropped into a square shape, with the side length equal to the shortest dimension of the original image, centered around its midpoint. Then a min-max image scaling was applied to the cropped satellite image prior to calving front prediction. The COBRA model predicts the entire coastline including both the fjord boundary (green line in 165 Figure 3b) and the glacier calving front (red line in Figure 3b) (Heidler et al., 2023), an example is shown in Figure 3b. Therefore, the model outputs need to be postprocessed to isolate the actual calving front.

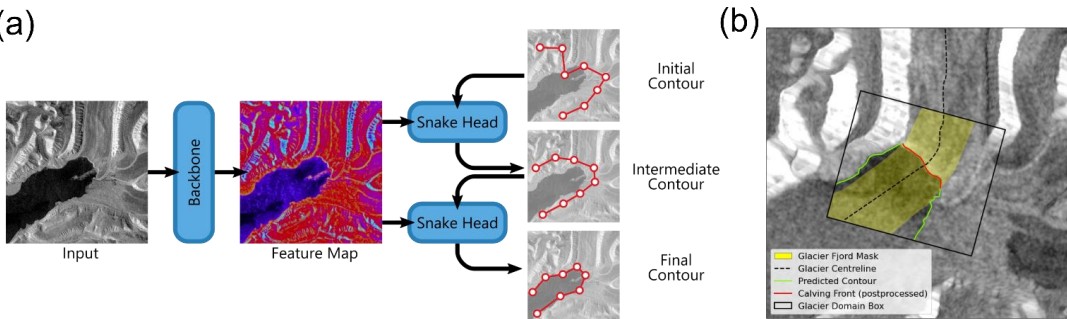

**Figure 3. (a) The Charting Outlines by Recurrent Adaptation (COBRA) deep learning model architecture used in this study (Heidler et al., 2023), here only shows two iterations of the snake head. (b) The calving front predicted by COBRA**
**model from Sentinel-1A SAR image on December 21, 2022 for Tunabreen glacier (RGI60-07.01458), the glacier fjord mask is shown as yellow polygon, the glacier centerline is shown as dashed black line, the model output is shown as**



**green line and red line, the postprocessed final calving front is shown as red line, the glacier domain box is shown as a black outline.**

### 2.3 Postprocessing

While deep learning techniques have demonstrated effectiveness in delineating glacier calving front locations (Cheng et al., 2021; Zhang et al., 2019; Heidler et al., 2022), many have only been trained on limited datasets, potentially missing some glacier terminus conditions in different satellite images. Consequently, due to the well-known distributional shift, the network may produce inaccurate predictions when processing satellite images that are not well-represented in the training datasets, e.g., where the calving front is less distinct, shadowing occurs, fast-ice is present or other factors. These inaccurate predictions need

to be removed from the final glacier calving front data product. In addition, the COBRA model prediction includes not only the glacier calving front, but also the neighbouring fjord boundary which is not needed. Here we developed an automated postprocessing pipeline to eliminate these inaccurate terminus traces and mask out the fjord boundary (Figure 4).

The pipeline consists of four major steps: 1) preliminary filtering of the initial COBRA model outputs based on the length and curvature of calving front line segments (Box 1 in Figure 4); 2) use of a fjord mask to exclude the fjord boundary or the other

non-calving-front features of each glacier (Box 2 in Figure 4); 3) identification and removal of erroneous traces based on glacier calving front line segment density and similarity (Box 3 in Figure 4); 4) utilizing a predefined glacier centreline to generate a time series of calving front changes and identifying outliers by applying a median filter to the calving front change times series (Box 4 in Figure 4).

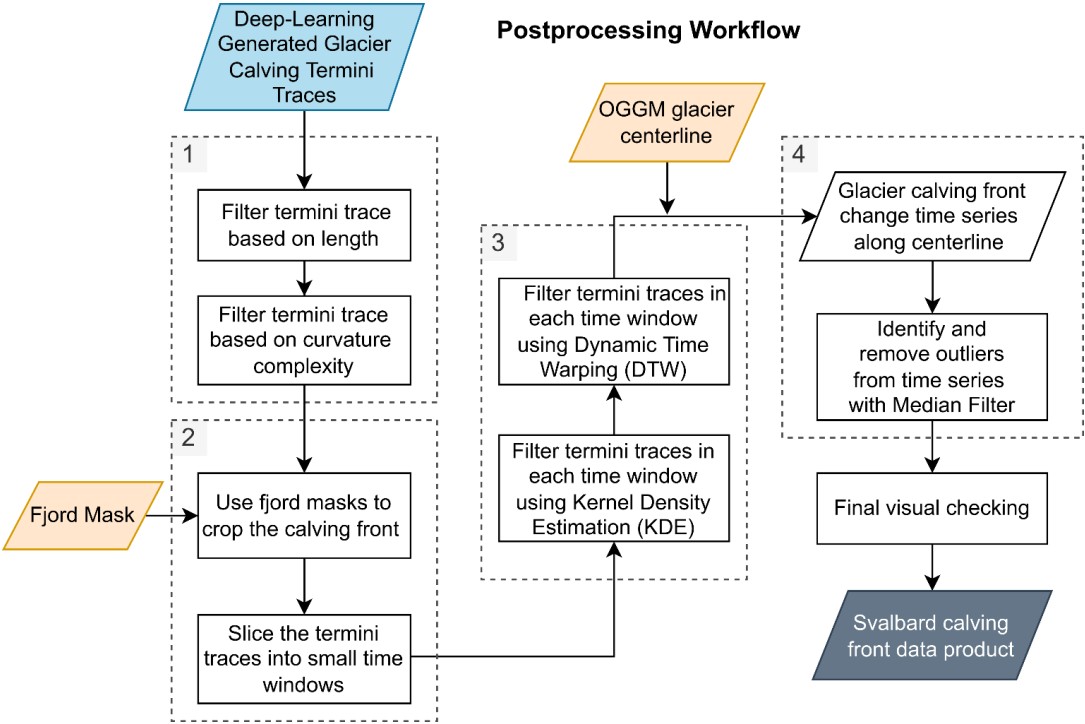


**Figure 4. The flowchart of postprocessing workflow applied to the glacier calving front traces mapped from the pretrained COBRA deep learning model.**



### 2.3.1 Filter original model output based on length and curvature

In cases where the glacier calving front is heavily obscured by cloud cover or high sea ice concentration, the calving front may
be less distinguishable in satellite images and the COBRA model can generate inaccurate predictions. These can manifest as either excessively short or long line segments and can exhibit overly complicated curvature shape. The first step of the postprocessing pipeline is to remove these inaccurate predictions according to the line segment length and curvature complexity (Box 1 in Figure 4). The terminus length and curvature filtering thresholds are based on the automatic screening module developed by Zhang et al. (2023). Two thresholds $T_L$ and $T_U$ based on the inter-quartile range were used for all the
initial terminus traces outputs from COBRA model in each glacier domain:

$$T_L = Q1 - 1.5 \times (Q3 - Q1) \quad (2)$$

$$T_U = Q3 + 1.5 \times (Q3 - Q1) \quad (3)$$

Where Q3 is the 75[th] percentile and Q1 is the 25[th] percentile of the data range. For the terminus length, we defined the terminus
traces from both the lower and upper thresholds $T_L$ and $T_U$ as outliers because the terminus traces either too short or too long
are likely to be anomalies. Following the length filtering of the terminus traces, we calculated the curvature of each terminus trace as the average for the curvatures between two adjacent points along each terminus trace, then eliminated the terminus traces with curvature values exceeding the upper threshold $T_U$. The reason of only applying an upper threshold for curvature complexity is because the high-quality terminus trace should be smooth with minimal curvature (Zhang et al., 2023).

### 2.3.2 Crop and filter glacier calving front using fjord mask

Following the initial filtering of terminus trace outputs based on the line segment length and curvature complexity in Section 2.3.1, a fjord mask was implemented for each glacier (yellow polygon in Figure 3b). As the model output includes both fjord boundary (i.e. land-water contact) and the glacier calving fronts, the fjord mask serves to exclude the fjord boundary, retaining only the calving front line segment that we are interested in (Box 2 in Figure 4). The fjord mask was generated by combining the ice-free zone from a binary ice mask and the land zone from a binary land mask.

The binary land mask was created using the high resolution (3"; ~90 m) Water Body Mask (WBM) product – showing inland water bodies and oceans – that is supplied with the Copernicus GLO-90 digital elevation model (DEM) dataset (European Space Agency (ESA), 2021). The WBM, together with the DEM product, is referenced on the WGS-84 ellipsoid and is provided in 1°x1° tiles globally. We used the RGI version 6 (RGI Consortium, 2017) first order region shapefile for Svalbard to compile the appropriate list of WBM tiles. After mosaicking all the WBM tiles for Svalbard, we converted the original
WBM product to a binary land mask by recategorizing all non-ocean pixels as land. The land mask mosaic was then reprojected to a 250 m grid (EPSG:3574) and clipped with the RGI region outlines. The binary ice mask was created using RGI version 6 glacier outlines for Svalbard. These are provided in shapefiles and then rasterised to the 250 m resolution land mask mosaic grid, which was applied to correct for any potential mismatches (i.e. masking out the ocean) between the RGI and Copernicus datasets.

After compiling the binary land and ice masks, we combined the two to find ice free land and vectorised the resulting product. As a final step we added a buffer zone of 200 m length to the merged ice-free land polygon then removed this buffered polygon from the glacier domain box to get the fjord mask that was used in subsequent steps (yellow polygon in Figure 3b). All the fjord masks for our glacier domains were visually checked and manually adjusted if necessary to make sure the mask can cover the entire calving front changes. The glacier calving fronts that have been clipped using fjord masks were subsequently
categorized into individual time windows which were defined based on the observation density. During the period of 1970 -



2015 when the data collection was limited, we set five distinct time windows: 1970-01-01 to 1990-01-01; 1990-01-01 to 2000-01-01; 2000-01-01 to 2005-01-01; 2005-01-01 to 2010-01-01; and 2010-01-01 to 2015-01-01. From January 2015 until January 2023, we set 17 time intervals, each spanning six months.

### 2.3.3 Terminus filtering based on the line segment density and similarity

Within a given time window defined in Section 2.3.2, we assume that the contour shapes of the majority of terminus traces are similar, and any erroneous terminus trace will significantly deviate from this expected similarity. Guided by this principle, we subsequently implemented two additional filtering steps for the clipped glacier calving front line segments: Kernel Density Estimation (KDE) and Dynamic Time Warping (DTW) (Box 3 in Figure 4). KDE is a well-established nonparametric approach to estimate the continuous density function based on a sample dataset and can cope with an inhomogeneous distribution of observations (Davies et al., 2018). Here it was used to estimate the density distribution of the glacier calving fronts. We firstly converted all the terminus trace line segments in one glacier domain into scattered points, then calculated their kernel-density estimates using a Gaussian Kernel. For the density map, we set the upper threshold as 75% percentile Q3 and extracted the contour boundary of the area where the KDE density is higher than Q3 – area inside this contour was taken as the boundary where glacier calving fronts are mostly likely to locate. For every terminus trace line segment, calculated its intersection with this Q3 contour. Termini traces situated completely outside the threshold contour were identified as outliers and subsequently excluded from the data product. Termini traces completely enclosed within the threshold contour, or those > 95% of the total trace length within the contour polygon, were taken as potential valid results and retained for subsequent postprocessing steps.

DTW is a technique that has been used in time series analysis to measure similarity between two sequences that vary in time and speed, and find the optimal alignment by accommodating time shifts and local shape distortions (Müller, 2007). Here we use DTW to measure the similarity between two different glacier termini trace line segments. For each terminus line segment, the DTW distances between this line segment and all the remaining terminus line segments were calculated. The resulting mean value was taken as the ultimate DTW distance for this terminus trace. After iterating this step for all the terminus traces within a given time window, an outlier detection threshold of 75% percentile Q3 of all the DTW distances was applied to identify the anomalous termini traces. If the DTW distance of a given terminus trace exceeds this threshold, it was eliminated from subsequent processing.

### 2.3.4 Calving front change time series and median filtering

The primary objective of measuring glacier calving front locations is to determine changes over time. Therefore, as a final step, we generated a calving front change time series for each glacier using a centreline approach and used this to remove outliers. The centreline approach measures the advance or retreat of the glacier calving front along a glacier centreline in relation to their earliest position (Cheng et al., 2021). The glacier centrelines for all the marine-terminating glaciers analysed in this study were first derived using the Open Global Glacier Model (OGGM) (Maussion et al., 2019). The OGGM glacier centreline was based on a predefined glacier domain boundary from RGI glacier database (Pfeffer et al., 2014), therefore its length may not cover all the calving front traces mapped in this study as some glaciers undergo dramatic changes at their calving fronts during the study period. To address this issue, we automatically extended the endpoint of each OGGM centreline by an additional 10 km in the seaward direction, following the direction defined by the line segment connecting the two outermost seaward data points of the OGGM centreline. In addition, only the main glacier centreline was extracted from OGGM model, for glacier domains located at the tributary glaciers we manually mapped the glacier centrelines. All the glacier centrelines were visually checked and modified when necessary to make sure it covers the entire glacier calving front locations of a given glacier and is near perpendicular to the calving front.

To make use of the dense glacier calving front observations after 2014, a rolling window of 10 observations was applied. We firstly calculated an upper threshold as the greater value between 200 m and the maximum standard deviation of calving front changes in all rolling windows. The range between the median calving front change distance in each rolling window above and below this threshold is served as the criteria for identifying and removing outliers. This assumes that within a short period of time with 10 observations, the glacier calving front change distance is likely to be less than 200 m (Luckman et al., 2015).

Furthermore, utilizing the highest standard deviation of calving front change observed across all rolling windows could accommodate the occurrence of large calving events. As a final step, all the glacier terminus traces after the above postprocessing steps were visually checked to make sure they are correct. The examples of different postprocessed glacier calving front traces for four different satellite sensors under different environmental conditions are shown as solid red lines in Figure 5.

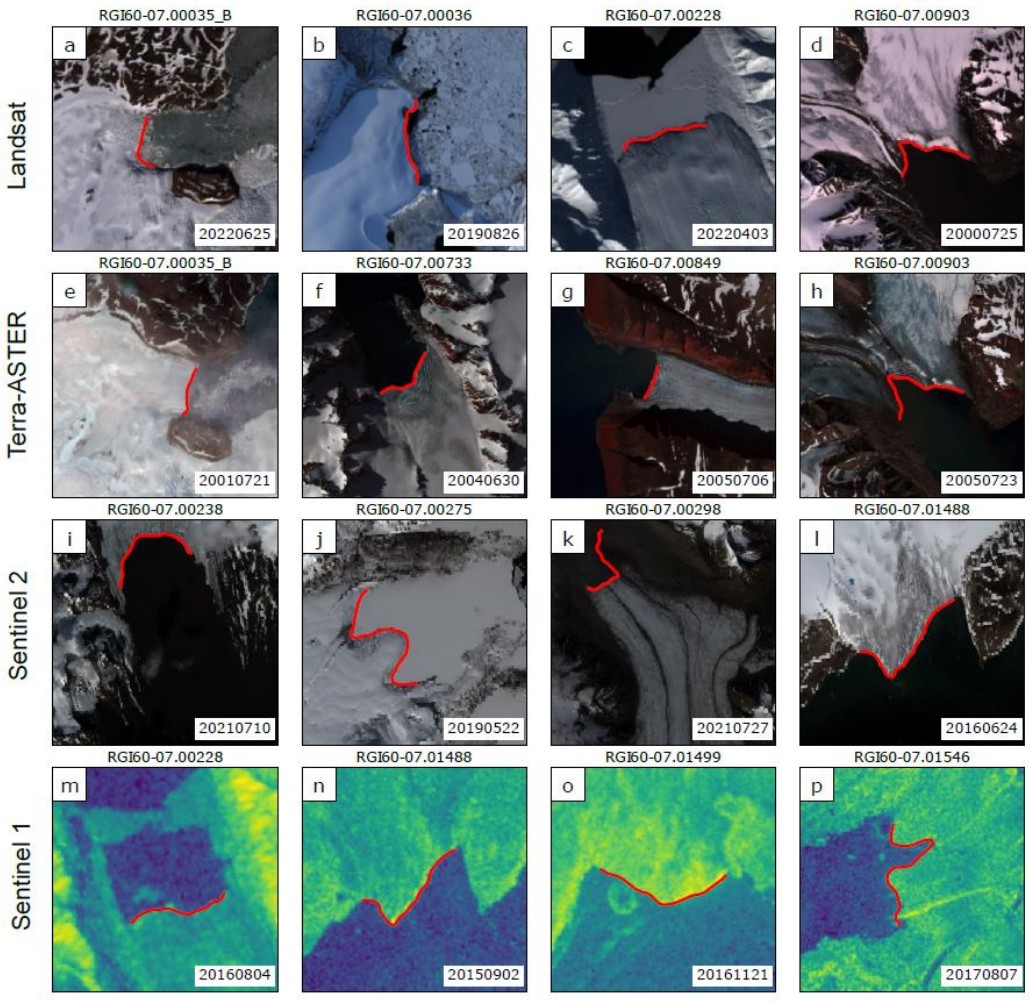


Figure 5. The examples of the postprocessed glacier calving front traces for different satellite images from four satellite platforms including Landsat (a-d), Terra-ASTER (e-h), Sentinel-2 (i-l) and Sentinel-1(m-p). The red solid lines are the final glacier termini traces after postprocessing.





## 3. Results

### 3.1 Dataset overview


Using the methodology developed in this study, we produced a new high-resolution calving front dataset which contains 124919 glacier calving fronts for 149 marine-terminating glaciers (based on updated glacier ids in Section 2.1) in Svalbard over the period of 1985-2023 (Li et al., 2023). The dataset is presented as a single GeoPackage file containing four different layers: the glacier calving front terminal traces mapped in this study, glacier centrelines generated in Section 2.3.4, glacier

domains generated in Section 2.3.2, and the along-centreline glacier calving front change time series in relation to the earliest time stamp. Each layer contains 149 different geometry features representing 149 marine-terminating glaciers. The detailed metadata provided in this GeoPackage file is shown in Table 2, including information on glacier id, satellite platform, satellite image id, satellite image acquisition date, and the glacier calving front change distance along the centreline. In addition, we also provided spatial distribution map plots of the glacier calving front traces and line plots depicting the time series of calving

front changes for each individual glacier. These plots are provided in PNG file format and can be accessed within the Figures folder.

The greatest number of traces were obtained after 2014 due to the availability of Sentinel-1 and Sentinel-2 satellites (Figures 6, 7 and A1), the low trace number in 2023 is because we only downloaded images in January. The annual average number of traces per glacier between 2014-2022 is 100, representing an average temporal resolution of 4 days. This allows us to discern

the seasonal patterns of glacier calving front changes. We demonstrate this in the case of five glaciers across Svalbard including a surging glacier Osbornebreen that exhibit strong seasonal signals after 2014 (Figure 7). A glacier's calving fronts normally retreat during the Arctic summer and autumn, and readvance during the Arctic winter and spring. The manually mapped areal change polygons of Kochtitzky and Copland (2022) only contain three different calving front traces for the years 2000, 2010 and 2020, thus this dataset cannot resolve any seasonal cycles or sudden changes in glacier calving front locations such as the

surging event shown in Figure 7l. However these polygons align well with our calving front traces (Figure 7b, e, h, k n).

**Table 2. Glacier calving front trace metadata recorded in the data product.**

| Data Field | Description |
| --- | --- |
| Glacier | The Randolph Glacier Inventory (RGI) version 6 glacier id. |
| Sensor | The satellite platform used in mapping glacier calving front, including "Landsat", "Terra-ASTER", "Sentinel2" and "Sentinel1". |
| ImageId | The image id of the satellite image used in mapping the glacier calving front. |
| DateString | The datetime string of the satellite image in the format of "YYYYMMDD". |
| CFL_Change | The calving front location (CFL) change in meters along the glacier centreline in relation to the earliest calving front location in the time series. |





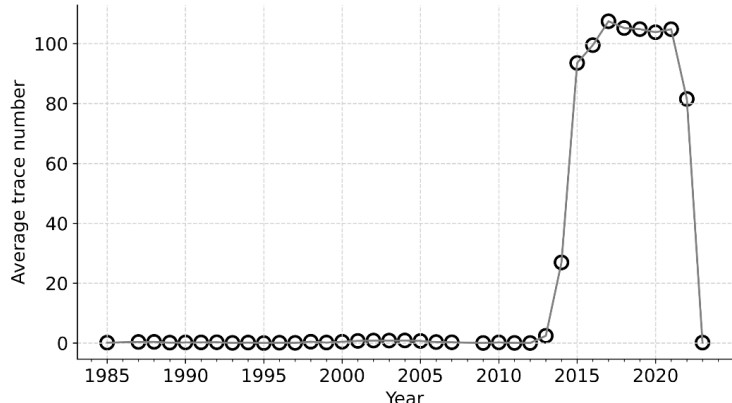

**Figure 6. Average calving front traces for all marine-terminating glaciers analyzed in this study from 1985 until 2023**

**January.**



(a) (b) (c)

(d) (e) (f)

(g) (h) (i)

(j) (k) (l)

(m) (n) (o)

**Areal Change Polygon** ⸱⸱⸱ Loss 2000-2010 ⸱⸱⸱ Loss 2010-2020 ⸱⸱⸱ Gain 2000-2010 ⸱⸱⸱ Gain 2010-2020

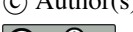



**Figure 7. Examples of glacier calving front change time series of five different glaciers located across Svalbard. Red dots in subplots (a), (d), (g), (j) and (m) show the locations of each glacier, the basemap is the S100 topographic raster data for Svalbard (https://data.npolar.no/dataset/44ca8c2a-22c2-49e8-a50b-972734f287e3, last accessed on April 17, 2023). In subplots (b), (e), (h), (k) and (n), coloured line segments are the glacier calving front traces mapped in this study for each glacier, they are overlaid with the 2000-2020 glacier areal change polygons (Kochtitzky and Copland, 2022) denoted by coloured dashed polygons (legend at the bottom of the figure), and the binary land-ice (white) and water mask (grey) generated in Section 2.3.2. Subplots (c), (f), (i), (l) and (o) show the glacier calving front change time series in relation to the earliest calving front trace at each glacier, blue crosses denote the calving front change observations before 2014. In subplot (l), the yellow polygon denotes the glacier surging event happened around 2020.**

### 3.2 Uncertainty and Validation

#### 3.2.1 Uncertainty measurement

The accuracy of the predicted calving front locations from the COBRA deep learning model depends on the spatial resolution of satellite images, the presence of cloud and shadow in optical images, speckle noises in SAR images, and the local sea ice conditions in front of the glacier terminus. The uncertainties related to the COBRA model have been evaluated by cross-validation on three different test datasets and by comparing with different deep learning models that were trained on the same training datasets, details can be found in Heidler et al. (2023). The average prediction error of COBRA on CALFIN test set is $99 \pm 10$ m. The rigorous postprocessing steps developed in Section 2 were able to eliminate the erroneous terminus trace predictions effectively. However, the measurement error still remains even after postprocessing and varies with different satellite images obtained on different time as the environmental conditions at the glacier calving front are different. To estimate the calving front mapping uncertainty in our final data product, we compare different terminal traces mapped in the same day for a given glacier by measuring the variation in their calving front locations along the glacier centreline. The average along-centreline calving front variation in days with multiple traces is then taken as the calving front mapping uncertainty of this glacier (Figure 8a). This is based on the hypothesis that calving front remains unchanged over a 24-hour period, and traces generated from different images during the same day should be the same. 85% of the evaluated glaciers have uncertainty less than 50 m (Figure 8b). On average, the calving front mapping uncertainty across Svalbard is $46 \pm 21$ m. Note the uncertainty measurement is sensitive to the centreline location. In our case, most of the centrelines are in areas where significant calving changes have been happening, therefore the estimated uncertainty should be representative for the glaciers analysed in our study.

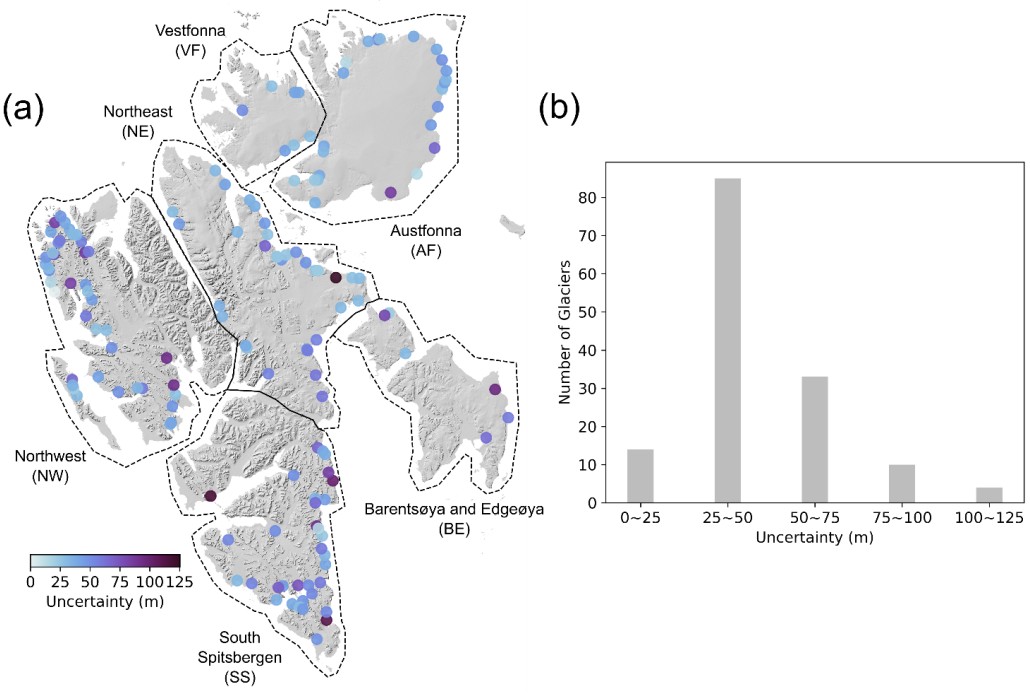


**Figure 8. Calving front mapping uncertainties for 146 glaciers (3 glaciers do not have duplicated traces on the same day). (a) Spatial distribution of calving front mapping uncertainty of different marine-terminating glaciers; (b) Histogram of different uncertainty categories.**

**3.2.2    Validation with another data product**

To assess the glacier calving front change time series produced in this study, we compared the long-term calving front change rate of each glacier with the Moholdt et al. (2022) annual glacier calving front data product as part of the Copernicus Glacier Service project. This product is the most complete glacier calving front data product for Svalbard prior to our study. It contains 12 years of calving front traces between 2008 and 2022 for 202 marine-terminating glaciers, and the total number of glacier 350 terminus traces is 2419 (Table 3). Moholdt et al. (2022) generated annual shapefiles of the marine-terminating glacier calving fronts by manual delineation from optical satellite imagery mainly available from Landsat-8 and Sentinel-2 during the period of August 15th to September 15th each year. We used the same centreline approach, with the same centrelines, to generate the glacier calving front change time series for the Moholdt et al. (2022) data product. Due to a mismatch in the marine-terminating glaciers included in these two different datasets, we analysed the common subset of 129 glaciers and compared their calving 355 front change rates.

Variations in observation densities over time among the glaciers in our dataset could introduce a potential bias in the linear regression analysis for estimating the long-term calving front change rates, which is not an issue for the Moholdt et al. (2022) data product with an annual temporal resolution. In order to facilitate the comparison of calving front change rates, we firstly converted the irregular calving front positions in our dataset to daily front change distances through linear interpolation, then 360 we calculated the monthly mean glacier calving front change distances. The calving front change rate was estimated by fitting a linear regression to the interpolated monthly front change time series. For each glacier, the calving front change rates were calculated within a common time window, which was defined by the overlapping time period between these two data products.

There is an excellent match between the spatial distribution of glacier calving front change rates obtained from the two products (Figure 9a-b). The glacier calving front change rates derived from this study show a significant near-linear correlation with the

glacier calving front change rates from Moholdt et al. (2022) ($R^2 = 0.98$, P-value < 0.05) (Figure 10a). The Morsnevbreen Glacier exhibits the highest advancing rate of around -700 m/yr in both products (Figure 10a). This glacier, known for its surging behaviour, experienced its most recent surging event between late 2016 and late 2018, during which it advanced approximately 5 kilometers (Figures A2a-c). At the Polakkbreen Glacier, the most significant calving front retreat rate is observed (Figure 10a). Over the period from 2016 to 2022, this glacier experienced a retreat of approximately 4 kilometers

(Figures A2d-f). In addition, 92% of the investigated glaciers show an absolute difference in calving front change rates less than 25 m/yr between the two data products (Figure 9c and Figure 10b). The Storisstraumen Glacier in Austfonna Basin-3 exhibits the largest absolute difference in front change rate of 77 m/yr (Figures A2g-i). Our data shows a pronounced seasonal cycle in the calving front change of this glacier during the period from 2014 to 2023 (black line in Figure A2i). In contrast, Moholdt et al. (2022)'s calving front measurements only record the most advanced location in September each year, resulting

in an underestimation of the calving front advancing rate.

**Table 3. Overview of two different calving front data products. "Type" indicates the type of calving front data is provided in the data product. "Method" indicates the how the dataset is produced. "No. glaciers" gives the number of presented glaciers. "No. mapped fronts" gives the total number of glacier calving front traces included in each data product.**

| Dataset | Data Source | Type | Method | No. glaciers | No. mapped fronts | Time span | Temporal resolution |
|---|---|---|---|---|---|---|---|
| Ours | Optical and SAR | Line | Neural Network | 149 | 124919 | 1985-2023 | Sub-weekly after 2014 |
| Moholdt et al. (2022) | Optical | Line | Manually | 202 | 2419 | 2008-2022 | Annually |


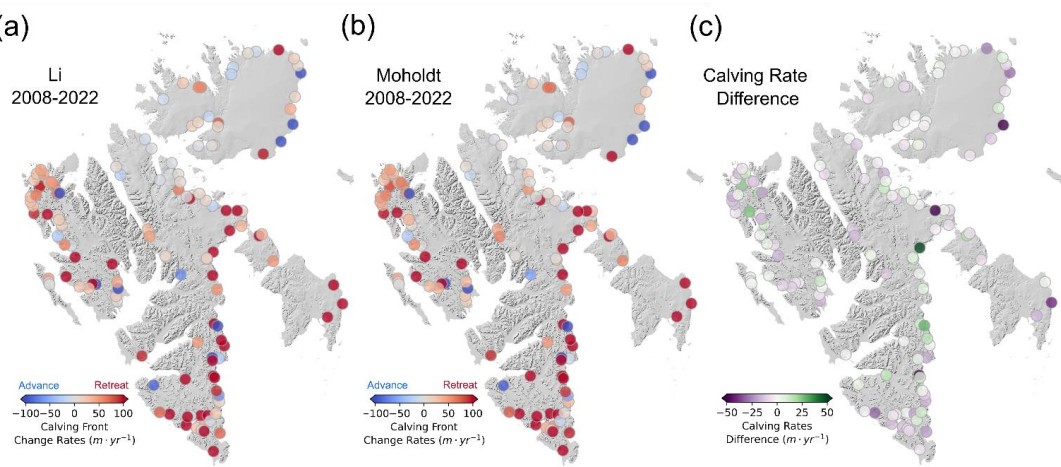

**Figure 9. The calving front change rates between 2008 and 2022 for the calving front data product generated in this study (a), calving front data product by Moholdt et al. (2022) (b), and the calving front change rate difference between these two calving front data products (c).**

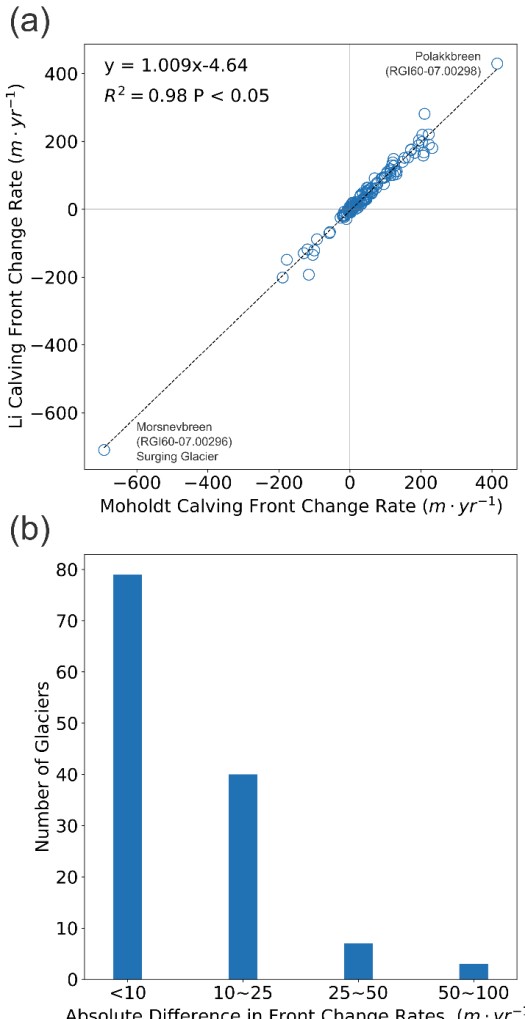

**Figure 10. Comparison of glacier calving front change rates between product generated in this study and the Moholdt et al. (2022) calving front data product. Subplot (a) shows the correlation between the glacier calving front change rates between these two different data products. Subplot (b) shows the histogram of absolute difference in glacier front change rates between these two different calving front data products.**

### 3.3 Spatial and Temporal Calving Front Variability in Svalbard

The spatial distribution of the different calving front change trends of the 149 marine-terminating glaciers included in the data product is shown in Figure 11. The predominant trend among Svalbard's marine-terminating glaciers is retreat, where 123 glaciers (82.6%) have been consistently retreating during the study period. 16 glaciers showed advancing trend (not surging), most of these glaciers are located on the Vestfonna and Austfonna ice caps on the island of Nordaustlanet at the noertheastern limit of the archipelago, where warm North Atlantic waters are less accessible (Figure 11) (Skogseth et al., 2005). There are an additional 10 glaciers that displayed surge behaviour and they have a widespread distribution across different regions.

Svalbard is one of the most prominent regions of surge-type glaciers, with approximately 13% showing this behaviour (Jiskoot et al., 2000). Using our extensive satellite data catalog, we were able to capture the exact timing of surge type events (Figures 8j-l and 12) and identify surging events that are unknown from previous calving front data products (Kochtitzky and Copland, 2022; Moholdt et al., 2022). For example, Tunabreen is a quiescent-phase surge-type glacier which terminates in Temperfjorden, a shallow fjord with limited connection to the warm ocean currents (Luckman et al., 2015). During our study period, we observed two individual surging events at Tunabreen, one in 2002-2004, and the other in 2017-2019 (orange boxes in Figure 12c). During both events, the Tunabreen calving front advanced more than 1.5 km in less than 2 years. In comparison, Moholdt et al. (2022) only identified the second surging event (Figure 12d), they were also unable to capture the seasonal cycles of the calving events. This example demonstrates the power of our highly automated multi-sensor calving front mapping scheme, which can uncover previously unknown events in unprecedented detail and can aid future investigations on calving front dynamics and the mass balance of tidewater glaciers.

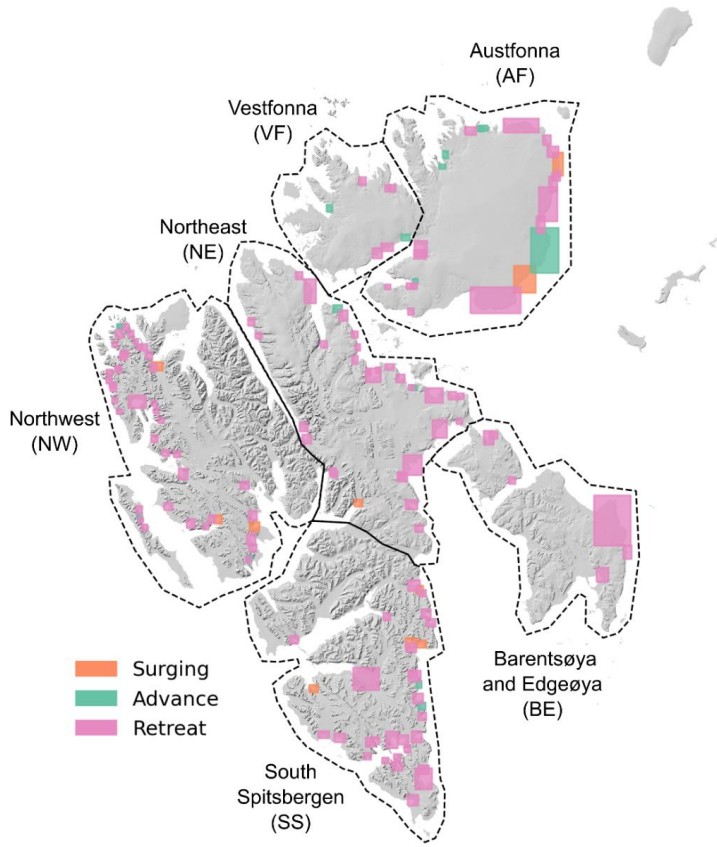

**Figure 11. Spatial distribution of different calving front change trends of marine-terminating glaciers in Svalbard derived from the calving front data product generated in this study, and the main current circulation around the Svalbard archipelago (Skogseth et al., 2005; Misund et al., 2016). The orange, green and pink polygons represent surging glaciers, non-surging type advancing glaciers and retreating glaciers, respectively.**

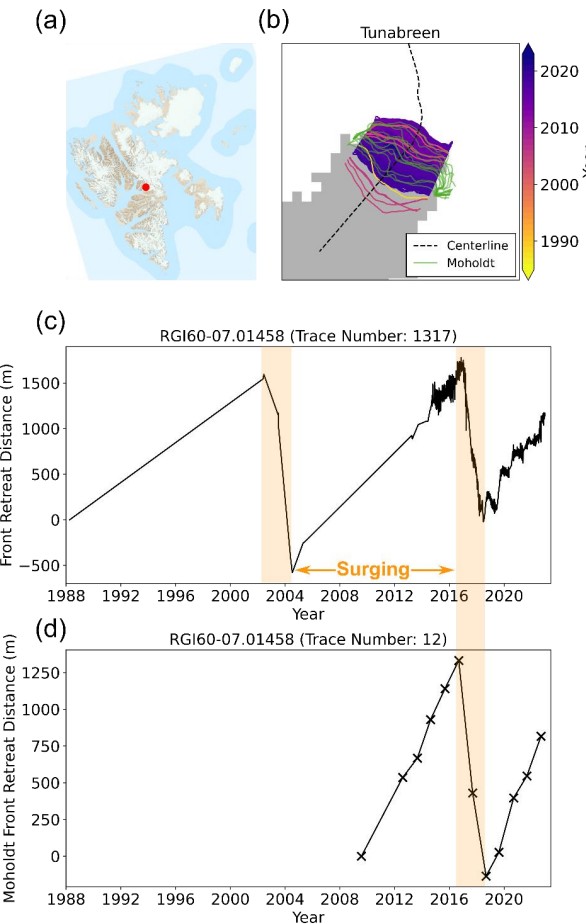

**Figure 12. Calving front change time series of Tunabreen surging glacier (RGI60-07.01458). (a) Red dot shows the location of Tunabreen overlaid on the basemap from the S100 topographic raster data for Svalbard (https://data.npolar.no/dataset/44ca8c2a-22c2-49e8-a50b-972734f287e3, last accessed on April 17, 2023); (b) The colored lines are the calving front traces derived in this study overlaid on the binary land-ice (white) and water mask (grey) generated in Section 2.3.2, the green solid lines are the calving front traces mapped in Moholdt et al. (2022) data product, the glacier centerline was denoted by dashed black line; (c) The glacier calving front change time series included in this study, the orange transparent boxes denote two individual surging events; (d) The glacier calving front change time series from Moholdt et al. (2022) data product, black crosses denote the calving front measurements.**

## 4. Discussion

Our calving front dataset of Svalbard marine-terminating glaciers in 1985-2023 is the first to provide calving front observations of large and comprehensive spatial coverage, high temporal resolution and a long time span of 38 years. It not only captures the spatial pattern of evolving marine-terminating glacier calving fronts, but also provides insights at different time scales. This dataset can be used to study glacier mass balance, understand calving mechanisms, and predict glacier dynamics.

The calving front data product is mapped using the novel COBRA deep learning model (Heidler et al., 2023). This model has been proven to outperform the previous calving front mapping models such as HED-Unet (Heidler et al., 2022) which was



used for the IceLine Antarctic ice shelf front dataset (Baumhoer et al., 2023), CALFIN (Cheng et al., 2021), as well as the
       UNet model (Mohajerani et al., 2019b). While the geomorphological features of tidewater glaciers in Svalbard and Greenland
       exhibit general similarities, it is important to note that the calving styles and neighbouring fjords can vary significantly among
       certain glaciers. Therefore, the CALFIN training dataset used in our model development may not be universally applicable to
       all Svalbard glaciers. To enhance the predictive capabilities of deep learning models and simplify post-processing procedures,
future research should focus on generating extensive training datasets for glacier calving fronts encompassing a wider range
       of geographical regions and glacier types.

       Several external datasets were needed as inputs for the preprocessing and postprocessing pipelines, including the Kochtitzky
       and Copland (2022) glacier front areal change polygon and the glacier centreline. The Kochtitzky and Copland (2022) areal
       change polygon serves the primary purpose of defining the glacier's bounding box for satellite image queries from GEE
platform. Given that this areal change polygon only covers a limited time period between 2000 and 2020, the fixed buffer
       length of 1.5 km used in Section 2.1 may not fully cover the entire calving front changes in 1985-2023. While this is less likely
       to be an issue in Svalbard given the relatively smaller scale and size of the marine-terminating glaciers, the buffer length will
       need to be adjusted when applying the processing pipeline to larger glaciers in different regions, such as the Greenland Ice
       Sheet. The glacier centreline is used in filtering out the abnormal front traces and producing the front change time series.
Although only one centreline is used for each glacier, the centrelines are placed in areas with substantial calving front changes,
       making it an effective and representative for filtering and quantifying the front changes over time.

       The calving front changes of marine-terminating glaciers in our study are consistent with earlier observations by Kochtitzky
       and Copland (2022) and Moholdt et al. (2022), though the temporal resolutions are different among these three products. The
       comparsion of our glacier calving rates with the Moholdt et al. (2022) annual calving front data product shows an excellent
match with $R^2 = 0.98$ during the time period of 2008-2022. The most significant mismatch in calving front change rate is
       located in Storisstraumen Glacier, this is due to Moholdt et al. (2022)'s annual calving front dataset fails to capture the seasonal
       calving cycles. This example demonstrates the importance of considering seasonal calving front changes when estimating the
       long-term front change rates. Both datasets exhibit a clear and predominant trend of glacier retreat across Svalbard, in
       agreement with the Kochtitzky and Copland (2022) study on decadal glacier calving front change during 2000-2020 which
shows that the net area change of glaciers in Svalbard is $-26.76 \pm 0.54\ km^2\ a^{-1}$. This spatial pattern was also reported by
       Geyman et al. (2022) by reconstructing DEMs using an archive of historical aerial imagery from 1936 and 1938, they show
       that the mass balance in Svalbard in 1936-2010 was dominantly negative with an average thinning rates of $0.35\ \pm 0.03\ m/yr$.
       Glaciers in most of the regions experienced thinning rates exceeding $0.5\ m/yr$, except the northeast Svalbard which remained
       stable during these 70 years.

Being able to assess calving front variability at multiple time scales is important in identifying drivers governing calving front
       changes and resolving mass balance estimations accurately (Benn and Åström, 2018; Rounce et al., 2023; Kochtitzky et al.,
       2022, 2023; Schuler et al., 2020; Luckman et al., 2015; Nuth et al., 2019; Strozzi et al., 2017; Cowton et al., 2018).
       Observations and theory show that increased calving can be driven by both atmospheric and oceanic warming. Increased
       surface melting and runoff can accelerate calving through hydrofracturing of near-terminus crevasses. In the meantime, it can
also increase subglacial discharge which, along with ocean warming, can drive submarine melting and accelerate teminus
       calving (Carr et al., 2013; Catania et al., 2020). Glacier calving processes in Svalbard, however, are not well understood due
       in part to a lack of comprehensive glacier calving front observations. Although Holmes et al. (2019) and Luckman et al.(2015)
       claimed that calving rates of marine-terminating glaciers in Svalbard vary strongly with ocean temperature, their results must
       be interpreted with caution – especially over large areas or long time-scales – as they only used a small sample of glaciers
(n<=3) within a short period of time of 2 years. The large number of investigated glaciers, along with the high temporal-





resolution and long time-span (1985-2023) of our data product provides a good basis to gain new insights into the governing mechanisms on marine-terminating calving processes in Svalbard.

**5. Code and data availability**

The source code of COBRA model v1.0.0 and inference examples are accessible at https://github.com/khdlr/COBRA/releases/tag/v1.0.0, its DOI is https://doi.org/10.5281/zenodo.8407566 (Heidler, 2023). The Svalbard calving front dataset produced in this study is available at the Zenodo data repository: https://doi.org/10.5281/zenodo.8399899 (Li et al., 2023).

**6. Conclusion**

In this study, we produced a new high-resolution glacier calving front dataset, including 124919 individual calving fronts, for
149 marine-terminating glaciers in Svalbard covering the period of 1985-2023. This represents a significant increase in glacier calving front observation density compared to similar products. This data product was derived using automated processing methods developed in this study, that incorporate a novel deep learning framework, multiple optical and SAR satellite images (Landsat, Terra-ASTER, Sentinel-1 and Sentinel-2) curated and downloaded via the Google Earth Engine platform, and a bespoke postprocessing algorithm. The data product is validated with the latest Svalbard annual calving front dataset produced
by Moholdt et al. (2022) by comparing the calving front change rates over the same period of time. The results show a strong correlation between the two products with an $R^2$ value of 0.98. Our results show that calving front retreat has been dominant across most of Svalbard in the past three decades, except the northeast region comprising of Vestfonna and Austfonna, consistent with the overall negative glacier mass balance identified in Svalbard. This new dataset will contribute to a better understanding of glacier calving front mechanisms and more accurate frontal ablation estimates in Svalbard, which is essential
in calculating glacier mass balance and predicting contribution to future sea-level rise especially in the context of the ongoing Arctic warming.

**Appendix A**

**Table A1. The Google Earth Engine (GEE) image collection for different satellite used in this study.**

| Satellite | GEE Image Collection |
|---|---|
| ASTER | ASTER/AST_L1T_003 |
| Landsat-1 | LANDSAT/LM01/C02/T1 |
| Landsat-2 | LANDSAT/LM02/C02/T1 |
| Landsat-3 | LANDSAT/LM03/C02/T1 |
| Landsat-4 | LANDSAT/LT04/C02/T1 |
| Landsat-5 | LANDSAT/LT05/C02/T1 |
| Landsat-7 | LANDSAT/LE07/C02/T1 |
| Landsat-8 | LANDSAT/LC08/C02/T1 |
| Landsat-9 | LANDSAT/LC09/C02/T1 |
| Sentinel-2 | COPERNICUS/S2_HARMONIZED |
| Sentinel-1 | COPERNICUS/S1_GRD |



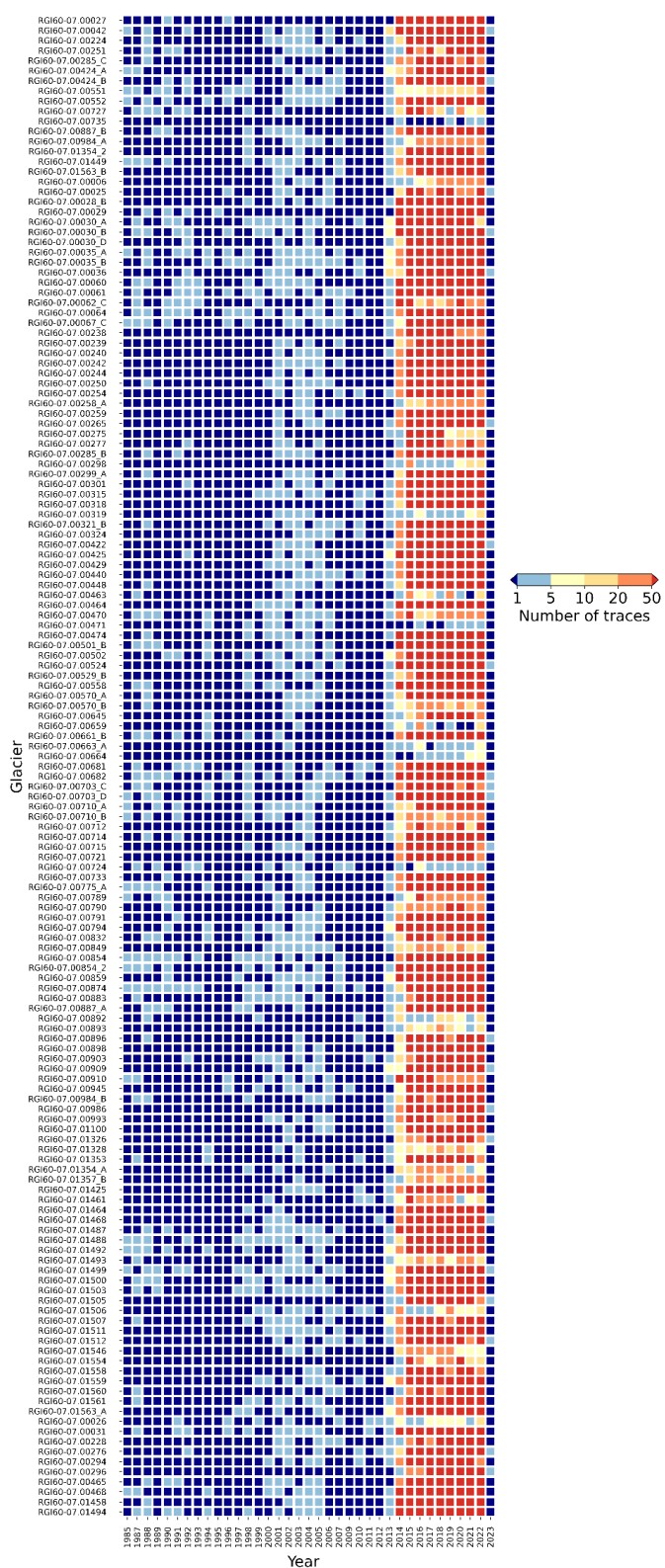

**Figure A1. Heatmap of glacier traces of each marine-terminating glacier analyzed in this study from 1985 until 2023**
**January. Each column represents one glacier, and each row represents one year ranging from 1985 to 2023. The color**
**corresponds to the number of traces for one glacier per year.**

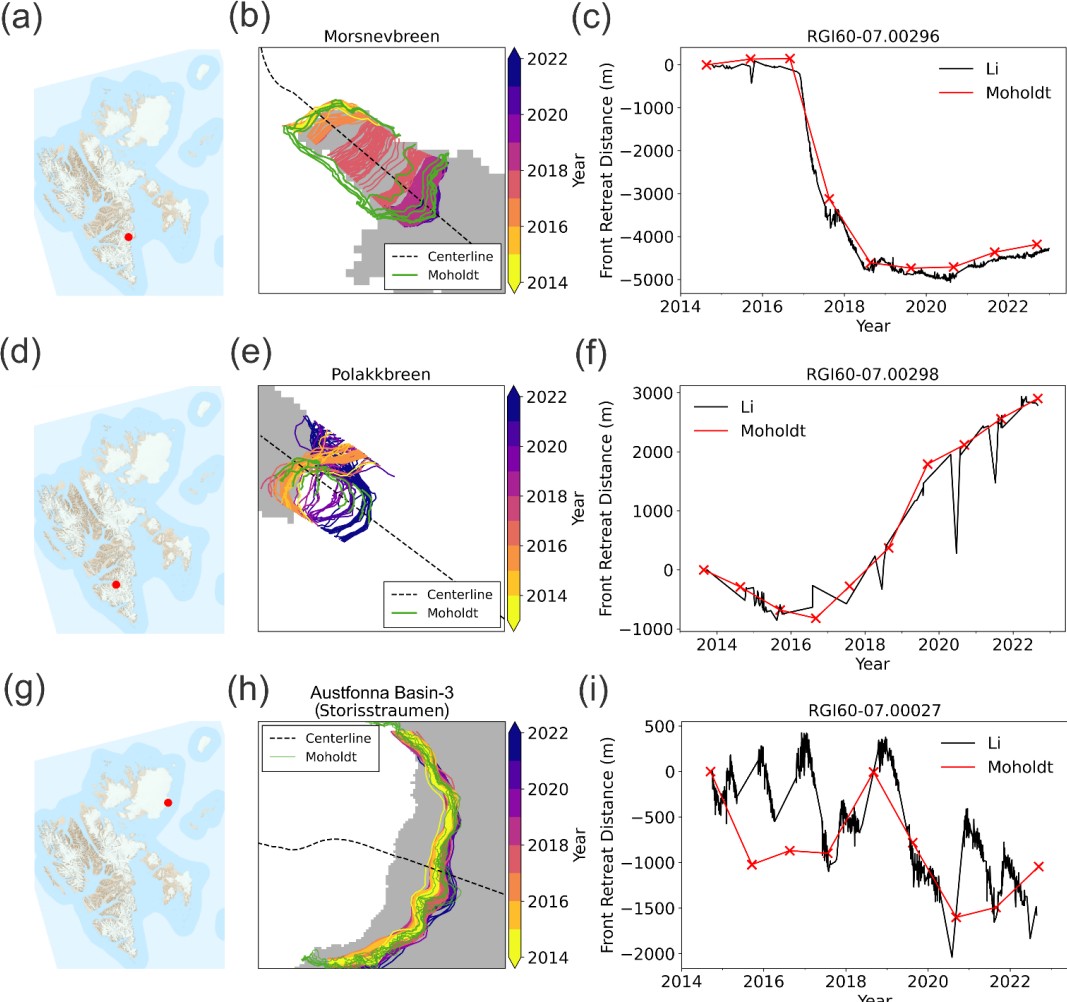

**Figure A2. Examples of glacier calving front change comparison during a common time period between calving front**
**data products generated in this study and by Moholdt et al. (2022) for Morsnevbreen Glacier (subplots a-c),**
**Polakkbreen Glacier (subplots d-f), and Storisstraumen Glacier in Austfonna Basin-3 (subplots g-i). Red dots in**
**subplots (a), (d), and (g) show the locations of each glacier, the basemap is the S100 topographic raster data for Svalbard**
**(https://data.npolar.no/dataset/44ca8c2a-22c2-49e8-a50b-972734f287e3, last accessed on April 17, 2023). In subplots (b),**
**(e), and (h), coloured line segments are the glacier calving front traces mapped in this study, they are overlaid with the**
**calving front traces mapped in Moholdt et al. (2022) denoted by green solid lines, and the binary land-ice (white) and**
**water mask (grey) generated in Section 2.3.2. Subplots (c), (f), and (i) show the glacier calving front change time series**
**in relation to the earliest calving front trace during the data comparison time window, black solid lines show the front**
**change time series generated in this study and the red solid lines show the Moholdt et al. (2022) front change time series.**

**Acknowledgement**



This work is primarily funded by the European Union's Horizon 2020 research and innovation programme through the project
        Arctic PASSION (grant number: 101003472). TL, LM and JLB also received funding from the German Federal Ministry of
        Education and Research (BMBF) in the framework of the international future AI lab "AI4EO -- Artificial Intelligence for Earth
        Observation: Reasoning, Uncertainties, Ethics and Beyond" (grant number: 01DD20001). KH received funding from German
        Federal Ministry for Economic Affairs and Climate Action in the framework of the "national center of excellence ML4Earth"
(grant number: 50EE2201C). XXZ received funding from the Munich Center for Machine Learning (MCML).

**Author contributions**

TL and JLB conceived the study. TL developed the data downloading, preprocessing and postprocessing workflows, produced
the results and wrote the paper. KH developed the deep learning model, trained the model and contributed to data preprocessing.
LM and AI contributed to the development of postprocessing pipeline. XXZ and JLB contributed to the interpretation of the
results. All authors commented on the manuscript.

**Competing interests**

The authors declare that they have no conflict of interest.



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
