# Peer review of "A High-Resolution Calving Front Data Product for Marine-Terminating Glaciers in Svalbard"

_Earth System Science Data, 2023_

## Author Comment (AC1)

We would like to thank the reviewers for their positive and constructive comments, please find our detailed responses to each comment presented by the reviewers in blue text below.

**Reviewer 01**

**General Comments:**

The work described in this publication details a glacial termini dataset for Svalbard, which covers 149 marine-terminating glaciers from 1985-2023, and is comprised of 124919 fully-delineated calving front positions. This data is generated using an automated processing pipeline to download optical (Landsat, Sentinel-2) and SAR (Sentinel-1, Terra-ASTER) remote sensing imagery via Google Earth Engine, and process the images into vectorized polylines using deep machine learning. This methodology is sound, building on and improving existing machine learning methods, which has been published alongside the data as the CORBA framework. The manuscript contains uncertainty and validation of the method/dataset, which provides appropriate bounds and checks on the accuracy and rigor of the automated methodology. The overall error is within human levels of accuracy of error ($46 \pm 21$ m $< 78$m, Goliber et al. 2022). Additionally, scientific analysis is performed on the dataset, showing calving front change in agreement with existing literature ($R^2 = 0.98$ during the time period of 2008-2022 with Moholdt et al., 2022) concerning the changes to Svalberd's marine-terminating glaciers.

The dataset itself consists of a single GeoPackage containing 4 outputs, which include the glacial centerlines, rectangular polygonal domains, front change time series/rates (in m/yr), and the calving front polyline traces. Metadata provides relevant information for reference and potential reproduction/reprocessing. It is well prepared, and in a common format that is easy for community members to use in future work. It is well documented both within the manuscript and in accompanying materials provided along with the dataset.

The publication is well done, and is largely free of grammatical errors and typographical issues. There are only minor remarks to be addressed by the authors, after which I can recommend acceptance at the editor's discretion.

Thank you very much for the positive comments on our study.

**Specific Comments:**

- For modelers and other community members, it would be useful to have the glacial termini data in the form of areal change or land/ocean polygonal masks, in addition to just polyline (as in Kochtitzky and Copland, 2022). This would involve connecting the existing calving front polyline endpoints to some static coastline that coincides with the boundaries of the polygonal domains already provided in the dataset. While this is not necessary to do and may be outside the scope of the publication, this would help reduce the processing required for community members to use the calving front data to mask ice extent changes in ice sheet models, or for measurement of areal change.

Thank you for the comment, this is a very good point. We agree that we have overlooked the importance of areal change polygons during our processing. We will keep this in mind in our future research on producing data products for other regions in the Arctic.

- Complementary with the previous comment, the glacial fjord mask is inexact, and enforces sharp cutoffs on the calving front endpoints where they join the fjord walls. While it may be out of scope to reprocess the dataset with improved fjord masks, this may prove useful to the community to avoid errors when connecting the calving fronts positions to land.

Thank you for the comment. When designing the experiments, we think the centerline-measured calving front changes are representative of the overall calving changes (King et al., 2020) and therefore have been focusing on developing automated methods for mapping calving front line segments, in align with previous deep-learning-focused studies (Baumhoer et al., 2023; Cheng et al., 2021; Loebel et al., 2023; Zhang et al., 2023). This is a very good point and as stated in our previous reply, we admit the importance of areal change polygons and we are confident that this issue can be solved in our future research by improving the fjord mask generation algorithms.

- For the glacial fjord boundaries, if it is possible or seen fit to include the glacial fjord masks in the final dataset, it might prove useful for further processing of the calving front traces. Again, this Is not strictly necessary if it is outside the scope of this work, but should increase the ease of using the data provided by this publication.

Thank you for the advice, we have now included the fjord masks in our final data product, please find the new version at https://zenodo.org/records/10407266.

- Figure 10 shows front rate change differences w.r.t. existing data, and Figure 12 shows a spatial distribution of calving front rate changes. In addition to these, it would be a good visual summary to show a histogram or distribution of calving front rate changes. Any relevant statistics (mean/median) would be good to include as well.

Thank you for the comment. According to the journal guideline (https://www.earth-system-science-data.net/about/aims_and_scope.html): "Articles in the data section may pertain to the planning, instrumentation, and execution of experiments or collection of data. Any interpretation of data is outside the scope of regular articles." Therefore, we would like to avoid too much scientific interpretation of the data and prefer not to include any additional statistical analysis of calving front change rates.

- While the standard error measurement seems to be the Area/Front length or Median Mean distance between predicted and ground truth fronts, this should be clarified in the text, as it may not be obvious how this is calculated without prior knowledge in the field.

Agree and done. We have now updated the error measurement by calculating the mean distance error (areal change normalized by the average front length) of the calving front traces mapped on the same day according to Reviewers 2 and 3. This has now been clarified in Section 3.2.1.

**Technical Comments:**

**Figure 8**: Consider adding more histogram bins for better granularity of the uncertainty/error distribution (i.e., 10m or 5m bins instead of 25m bins).

Agree and done, we have now changed the bin size to 10 m shown in Figure 8c (please see the figure in our reply to Reviewer 3).

**Figures 3, 9, 10**: Some text is small/hard to read – consider increasing the font size in these plots.

Agree and done, we have increased the font size in these three figures, note Figures 9 and 10 are now Figures 10 and 11.

**Reviewer 02**

**General Comments**

This paper presents a comprehensive dataset, generating 124,919 glacier termini for 149 marine-terminating glaciers in Svalbard. Employing an innovative automated deep learning pipeline, the dataset integrates multiple optical and SAR satellite images to enhance temporal coverage. The pipeline encompasses a GEE-based automated data collection method, the CORBA deep learning framework, and a suite of post-processing techniques designed to filter out inaccuracies, ensuring the integrity of the dataset.

Overall, the paper is well written, the method is solid, and the dataset is of high quality. The dense calving fronts will benefit future scientific research about the estimation of glacier mass loss and calving mechanism. Nonetheless, there are specific comments that require attention before acceptance, at the editor's discretion, as outlined below.

Thank you very much for the positive comments on our study.

**Specific Comments**

Line 108: Why does the number of marine-terminating glaciers listed here (220) differ from the 149 mentioned in the abstract?

The reasons of only 149 glaciers are available in the final product despite 220 glacier domain files were used in processing are:

1) Glaciers became land-terminating during the study period and were removed as a result;

2) Certain glaciers have a small amount of calving front traces (limited by the availability of limited satellite images), and they were identified as erroneous traces in the postprocessing pipeline due to strict filtering strategies.

Line 115: Did the author merge images captured on the same date before applying a non-data pixel threshold? The satellite stripe footage might cover only a portion of a glacier, but merging it with its adjacent stripe could provide complete images. This is generally applicable to optical images but not as much for SAR images. Therefore, combining images from the same satellite on the same date before applying the non-data threshold might yield a greater number of images.

Thank you for the comment, this is a good point. We did not merge the downloaded same-day satellite images before calving front prediction. We agree that merging incomplete satellite images acquired on the same day could provide a greater number of images. However, this will make the preprocessing pipeline excessively complicated, especially for SAR images as mentioned by the reviewer. Considering the large amount satellite images available, we decided to directly feed the GEE-downloaded images to deep-learning model without any merging steps, because even without these steps we could still obtain a high-density calving front data product as

demonstrated in our study, this also simplifies the whole processing workflow. We have clarified this point in Line 118-119: "In addition, we did not merge satellite images acquired on the same day."

Line 117: A total of 1,135,074 satellite images were downloaded, yielding 124,919 glacier termini. This suggests an abandonment rate of approximately 90%. I am concerned about this aspect because the utilization of the inter-quartile range in results filtering relies on the assumption that the majority of the results are of high quality.

Thank you for the comments. As requested by the Reviewer 3, we have calculated the number of terminus traces identified by the automated postprocessing steps to be 206371 (Line 281). The low ratio of 12% between successful calving front delineations and the total input satellite images can be possibly attributed to the following three reasons:

1) Downloaded satellite images may only cover an incomplete portion of the glacier calving front since we did not merge the images acquired on the same date, therefore they cannot provide successful calving front delineation;

2) Filtering using inter-quartile ranges indeed relies on the assumption that the majority of the results are correct, this assumption has been used throughout our postprocessing workflow in filtering the erroneous traces (Section 2.3). Please note that, instead of only using one inter-quartile range filtering, we have implemented four different steps that involve the inter-quartile range filtering, including length, curvature complexity, DTW and KDE filterings. These stacked processing steps could significantly reduce the number of final outputs;

3) The availability of a large input satellite image catalogue in our study allows us to implement a rigorous strict postprocessing workflow to keep the most confident terminus delineation while still retaining a high density. The whole point is to maximumly reduce the manual intervention to guarantee the quality of the final data product, considering the COBRA model is only trained on a limited training dataset from Greenland tidewater glaciers.

Line 152: The test error for CALFIN test set is 99 ± 10 m. What about the Baumhoer dataset?

The test error for the Baumhoer dataset is 99 ± 12 m (Heidler et al., 2023). We have clarified this in Line 334-335: "The average prediction error of COBRA on CALFIN test set is 99 ± 10 m, while it is 99 ± 12 m for the Baumhoer dataset (Heidler et al., 2023)."

As a suggestion for the post-processing step, I recommend applying 2.3.2 first, followed by 2.3.1. The rationale behind this recommendation is that a result might be accurate on the glacier terminus but incorrect on the fjord boundary, and we can still make use of this result. Applying 2.3.2 first would allow for the retention of such results, while applying 2.3.1 first might discard them. It's worth noting that this is a suggestion for the authors to consider, rather than a strict requirement.

Thank you for the suggestion. The reason of applying length and curvature filters first before clipping calving fronts with fjord masks is because if clipping calving front traces first with a fjord mask, this will result in erroneous randomly oriented calving front predictions (often with excessively short/long length and high curvature complexity) still kept inside the mask. When applying length and curvature filters to these calving fronts, it is possible that high-quality front

traces are filtered if the majority of clipped front traces are of low quality, especially for small glaciers with complicated surface morphology that is not well represented in the CALFIN training dataset.

Line 270: What about before 2014?

We did not apply rolling window for observations before 2014 due to lack of sufficient terminus traces because the available terminal traces number within one year could be less than 10.

Line 274: The full-thickness calving that produces tabular icebergs could cause glacier terminus to retreat by several kilometers within several days.

Good point. Large calving events mostly are stochastic events that are difficult to capture automatically, here we try to consider this possibility by implementing "an upper threshold as the greater value between 200 m and the maximum standard deviation of calving front changes in all rolling windows." (Line 273-274). Although this threshold is not perfect, tabular icebergs are less likely to be an issue for Svalbard, it will need to be further improved when applying the algorithm to tidewater glaciers in Greenland Ice Sheet.

Line 277: I assume the visual checking is performed manually, potentially impacting the pipeline's level of automation. How many incorrect calving fronts are identified through this visual checking?

We have identified 81452 incorrect calving fronts through visual checking, we have now mentioned this in Line 281: "206371 glacier calving fronts were identified by the automated postprocessing steps and 81452 terminus traces were discarded in the visual checking".

Figure 7: Is retreating symbolized by an upward trend or a downward trend? I suppose both (b) and (e) experienced retreating until recent years, but (c) has an upward trend while (d) has a downward trend. Please consider adding labels to show the retreat in all the figures of terminus variation.

Thank you for the comment. The upward trend denotes retreating while downward trend represents advancing, the positive retreat distance means the calving front has retreated compared to the earliest time stamp in a time series. We have now added an illustration inside Figure 7c to explain the retreating and advancing trends. The overall calving front migration trends for Austfonna Basin 3 (Figure 7b) and Austre Torellbreen (Figure 7e) are advancing and retreating, respectively.

Line 327: The test error for the Baumhoer dataset is missing.

Done. We have added the test error for the Baumhoer dataset in Line 334-335: "The average prediction error of COBRA on CALFIN test set is 99 ± 10 m, while it is 99 ± 12 m for the Baumhoer dataset (Heidler et al., 2023)."

Line 331: If I understand correctly, the uncertainty is assessed by quantifying the variance in model-predicted terminus traces from different satellite images captured on the same date. Is that accurate? If so, I recommend utilizing the test error to represent the dataset error and suggest corresponding adjustments to the abstract. Additionally, could you provide insight into how to deal with the duplicate results originating from different satellites on the same date?

Yes, in our preprint we calculated the uncertainty by quantifying the variance in terminus traces delineated on the same day along a centreline.

Based on the comment by Reviewer 3: "*Furthermore, the accuracy is calculated along one centerline not accounting for errors at the margins of the glaciers. That makes it difficult to compare the accuracy to other studies as they measured the accuracy by the mean difference between two fronts. Please consider providing accuracy measures of the mean distance error considering the entire front to make your accuracy assessment comparable to other studies.*", we now recalculated the product uncertainty by measuring the mean distance error between terminus traces obtained on the same day (Cheng et al., 2021; Loebel et al., 2023). The mean distance error now considers the entire calving front instead of just focusing on the interaction with glacier centreline. The average mean distance error across Svalbard is $31 \pm 30$ m. We think this mean distance error is a better representation for the overall product quality compared to the model test error, therefore we used this number in the abstract.

We do not expect duplicated results originating from different satellites on the same date deviate significantly from each other since the average mean distance error is small, therefore they should not be an issue in time-series analysis as this deviation will not alter the trend at a longer time scale. However, if users would like to have only one terminus trace for a single date, we suggest first connecting and merging different duplicated traces into one areal change polygon, then sampling the centerline of this polygon at a certain interval along the direction of fjord width.

Section 3.2.2 Some components of this section, such as the dataset used for result validation and the methodology employed for validation, may be more appropriately placed within the Methods section. I recommend restructuring this section to ensure a clearer delineation of content.

Thank you for this comment. We think it is better to keep as it is because Moholdt dataset and the associated method are for data validation only in this section, while in Methods Section we aim to focus on the general product processing framework.

Section 3.2.2: A more critical question arises regarding the necessity of using the rate of terminus changes for result validation. It seems more straightforward to calculate the difference between the results of this study and Moholdt et al. (2022) data on the same date, similar to the author's approach for quantifying uncertainty and test error. Such a direct comparison could serve as an additional test error, complementing the other two (CALFIN and Baumhoer test errors) to more comprehensively represent the error in this study's results.

Good point, thank you for the comment. We have now calculated the mean distance error to be $32 \pm 65$ m based on terminus traces obtained on the same day from Moholdt dataset and our data product (please also refer to the comment by Reviewer 3). The per-glacier mean distance error is shown below (now Figure 9). Given only 85 glaciers have the same-day calving front traces between these two data products, we decided to keep the comparison of terminus change rates for all the glaciers, as we think it provides valuable complementary information regarding product quality assessment.

[Figure]

Figure 9. Calving front mapping mean distance error for 85 glaciers between data products generated in this study and by Moholdt et al. (2022). (a) Spatial distribution of calving front mapping mean distance error of different marine-terminating glaciers; (b) Temporal distribution of the same-day calving front trace duplicates; (c) Histogram of different mean distance error categories.

Figure 12: The arrows in (a) can be misleading. It could make readers think the white period within the two arrows is the surging period.

Agree and done, we have now removed the arrows in this figure (now Figure 13).

**Reviewer 03**

**General Comments:**

Li et al. present a calving front dataset for 149 marine-terminating glaciers in Svalbard over the time span 1985-2023. The fronts were extracted from different satellite missions (Sentinel-1/2, Terra-ASTER, Landsat) available at Google Earth Engine (GEE). The fronts are extracted by the novel COBRA model based on a CNN and active contour model in contrast to previous U-Net-based approaches. The mapping accuracy along centerlines is very high (46 ± 21 m) and the calculated calving front change rates are consistent with existing studies ($R^2$ of 0.98) based on

manual front delineations. The created dataset is freely available and uses well-known RGI glacier domains making it a valuable resource for glaciological studies on calving front dynamics and ice-ocean interactions.

The manuscript is of high-quality, very well written and accompanied by good graphics. Only a few minor remarks should be addressed by the authors to improve the manuscript:

Thank you very much for the positive and constructive comments on our study.

**Specific Comments:**

L70: Are the short acquisition intervals of 1-3 days possible year-round or only in summer when polar night is not present?

It is possible all year around as Sentinel-1 can provide dense observations even in polar night.

L93: Why did you chose to use Sentinel-1 EW data and not the higher resolution IW data? For the glaciers in Svalbard I would assume the higher resolution would be very beneficial.

The reason we chose to use Sentinel-1 EW data is there exists a higher amount of EW data for Svalbard than IW data. Using Planetary Computer , we calculated the numbers of available EW and IW scenes for Svalbard to be 18183 and 3903, respectively. This has now been clarified in Line 95-96: "The reason of using the EW mode for Sentinel-1 images instead of the higher-resolution Interferometric Wide (IW) mode is that EW mode has a higher amount of SAR images available in Svalbard according to Planetary Computer".

Section 2.3.4 Excluding erroneous calving front detections has been a challenging task for a while and different approaches were developed such as the automated screening module based on front geometries (Zhang et al. 2023) or a time series approach (Baumhoer et al. 2023). Could you provide some numbers on how many fronts were identified by the automated approach and how many were removed by visual inspection? Additionally, it would be very beneficial to state the ration between successful front delineations and removed ones based on the number of available input images. Are these ratios more or less the same or are there fronts where COBRA preforms better/worse and less/more fronts are excluded?

The automated postprocessing pipeline identified 206371 terminus traces, 81452 were removed by visual inspection. The ratio of successful calving front delineations (124919) compared to all the input satellite images (1135074) is 12% (Line 282-284). Please also see our reply to Reviewer 2.

Section 3.2.1 The uncertainty measurement is based on fronts delineated on the same day from different image sources. This is for sure a good approach as it does not require tedious manual delineations. Nevertheless, it should be kept in mind, that several front positions at the same day are more likely available since the launch of Landsat-8/9 and Sentinel-1/2. Could you provide a time line to see the temporal distribution of your accuracy assessment? Does the accuracy assessment cover the entire time series between 1985 to 2023 or is it temporally clustered? Furthermore, the accuracy is calculated along one centerline not accounting for errors at the margins of the glaciers. That makes it difficult to compare the accuracy to other studies as they measured the accuracy by the mean difference between two fronts. Please consider providing accuracy measures of the mean distance error considering the entire front to make your accuracy assessment comparable to other studies (Cheng et al 2021, Loebel et al. 2023).

Thank you very much for this advice.

1) Yes the front positions mapped on the same day are temporally clustered in 2013-2022, shown in the Figure 8b below.

2) We now calculated the mean distance error as uncertainty. The average mean distance error across Svalbard is $31 \pm 30$ m, the per-glacier mean distance error distribution and its histogram are shown in Figure 8a and 8b. Details can be found in Line 339-349, Section 3.2.1.

[Figure]

Figure 8. Calving front mapping mean distance error for 146 glaciers (3 glaciers do not have duplicated traces on the same day). (a) Spatial distribution of calving front mapping mean distance error of different marine-terminating glaciers; (b) Temporal distribution of the same-day calving front trace duplicates; (c) Histogram of different mean distance error categories.

Section 3.2.2 Why did you decide to compare the rates and not the front positions itself? It would be very interesting to provide a mean distance error also for the Moholdt et al. 2022 dataset. For sure, also manual front delineations are not 100% accurate but the comparison would give a good estimate for general deviations that have to be considered. Also see the comment above on accuracy measures with the mean distance error.

Thank you for this comment. We have now compared the front position itself between Moholdt dataset and our data product in Line 361-371. The mean distance error between these two data products is $32 \pm 65$ m based on terminus traces obtained on the same day. The per-glacier mean

distance error distribution is shown in Figure 9, please refer to our reply to Reviewer 2. Only 85 glaciers have the same-day calving front traces between these two data products, therefore, we decided to keep the comparison of terminus change rates for all the glaciers as we think this comparison still provides valuable complementary information regarding product quality assessment.

Figure 12 (b): Nice exemplary figure but unfortunately it is hard to see the Moholdt et al. fronts below the purple fronts of your dataset. Additionally, you could provide a magnified part of the figure in the area where fronts are overlapping.

Agree, we have now added a zoomed-in figure to show the details (now Figure 13).

[Figure]

Figure 13. Calving front change time series of Tunabreen surging glacier (RGI60-07.01458).

Figure 12 (c/d): It is a bit misleading that the retreat distance is positive and the advance negative. I would recommend to flip the y-axis and display frontal change where a surge is positive as the front advances and retreat is negative.

Thank you for pointing this out, we have now added an illustration in Figure 13e (and Figure 7c) to show the retreating (upward trend) and advancing (downward trend) directions of calving front migrations.

References mentioned in this review:

Baumhoer, C. et al. (2023): IceLines – A new data set of Antarctic ice shelf front positions, Scientific Data, 10(1), p. 138. Available at: https://doi.org/10.1038/s41597-023-02045-x

Cheng, D. et al. (2021):Calving Front Machine (CALFIN): glacial termini dataset and automated deep learning extraction method for Greenland, 1972–2019, The Cryosphere, 15(3), pp. 1663–1675. Available at: https://doi.org/10.5194/tc-15-1663-2021.

Loebel, E. et al. (2023): Calving front monitoring at sub-seasonal resolution: a deep learning application to Greenland glaciers, The Cryosphere Discussions, pp. 1–21. Available at: https://doi.org/10.5194/tc-2023-52.

Moholdt, G., Maton, J., Majerska, M., and Kohler, J. (2022): Annual frontlines of marine-terminating glaciers on Svalbard, https://data.npolar.no/dataset/d60a919a-9cc8-4048-9686-df81bfdc2338.

Zhang, E., Catania, G. and Trugman, D.T. (2023): AutoTerm: an automated pipeline for glacier terminus extraction using machine learning and a "big data" repository of Greenland glacier termini, The Cryosphere, 17(8), pp. 3485–3503. Available at: https://doi.org/10.5194/tc-17-3485-2023.

**References in Authors' Comments:**

Baumhoer, C. A., Dietz, A. J., Heidler, K. and Kuenzer, C.: IceLines – A new data set of Antarctic ice shelf front positions, Sci. Data, 10(1), 1–10, doi:10.1038/s41597-023-02045-x, 2023.

Cheng, D., Hayes, W., Larour, E., Mohajerani, Y., Wood, M., Velicogna, I. and Rignot, E.: Calving front machine (CALFIN): Glacial termini dataset and automated deep learning extraction method for Greenland, 1972-2019, Cryosph., 15(3), 1663–1675, doi:10.5194/tc-15-1663-2021, 2021.

Heidler, K., Mou, L., Loebel, E., Scheinert, M., Lefèvre, S. and Zhu, X. X.: A Deep Active Contour Model for Delineating Glacier Calving Fronts, IEEE Trans. Geosci. Remote Sens., 61, 1–12, doi:10.1109/TGRS.2023.3296539, 2023.

King, M. D., Howat, I. M., Candela, S. G., Noh, M. J., Jeong, S., Noël, B. P. Y., van den Broeke, M. R., Wouters, B. and Negrete, A.: Dynamic ice loss from the Greenland Ice Sheet driven by sustained glacier retreat, Commun. Earth Environ., 1(1), 1–7, doi:10.1038/s43247-020-0001-2, 2020.

Loebel, E., Scheinert, M., Horwath, M., Humbert, A., Sohn, J., Heidler, K., Liebezeit, C. and Zhu, X. X.: Calving front monitoring at sub-seasonal resolution: a deep learning application to Greenland glaciers, Cryosph. Discuss., 1–21, doi:10.5194/TC-2023-52, 2023.

Moholdt, G., Maton, J., Majerska, M. and Kohler, J.: Annual frontlines of marine-terminating glaciers on Svalbard, [online] Available from: https://data.npolar.no/dataset/d60a919a-9cc8-4048-9686-df81bfdc2338 (Accessed 9 August 2023), 2022.

Zhang, E., Catania, G. and Trugman, D. T.: AutoTerm: an automated pipeline for glacier terminus extraction using machine learning and a "big data" repository of Greenland glacier termini, Cryosph., 17(8), 3485–3503, doi:10.5194/TC-17-3485-2023, 2023.